# ARMCMC: Online Model Parameters full probability Estimation in Bayesian Paradigm

## Abstract

Although the Bayesian paradigm provides a rigorous framework to estimate the full probability distribution over unknown parameters, its online implementation can be challenging due to heavy computational costs. This paper proposes Adaptive Recursive Markov Chain Monte Carlo (ARMCMC) which estimates full probability density of model parameters while alleviating shortcomings of conventional online approaches. These shortcomings include: being solely able to account for Gaussian noise, being applicable to systems with linear in the parameters (LIP) constraint, or having requirements on persistence excitation (PE). In ARMCMC, we propose a variable jump distribution, which depends on a temporal forgetting factor. This allows one to adjust the trade-off between exploitation and exploration, depending on whether there is an abrupt change to the parameter being estimated. We prove that ARMCMC requires fewer samples to achieve the same precision and reliability compared to conventional MCMC approaches. We demonstrate our approach on two challenging benchmark: the estimation of parameters in a soft bending actuator and the Hunt-Crossley dynamic model. Our method shows at-least 70% improvement in parameter point estimation accuracy and approximately 55% reduction in tracking error of the value of interest compared to recursive least squares and conventional MCMC.

## 1 Introduction

Bayesian methods are powerful tools to not only obtain a numerical estimate of a parameter but also to give a measure of confidence (Kuśmierczyk et al., 2019; Bishop, 2006; Joho et al., 2013). In particular, Bayesian inferences calculate the probability distribution of parameters rather than a point estimate, which is prevalent in frequentist paradigms (Tobar, 2018). One of the main advantages of probabilistic frameworks is that they enable decision making under uncertainty (Noormohammadi-Asl & Taghirad, 2019). In addition, knowledge fusion is significantly facilitated in probabilistic frameworks; different sources of data or observations can be combined according to their level of certainty in a principled manner (Agand & Shoorehdeli, 2019). Nonetheless, Bayesian inferences require high computational effort for obtaining the whole probability distribution and prior general knowledge on noise distribution before estimation.

One of the most effective methods for Bayesian inferences is the Markov Chain Monte Carlo (MCMC) methods. In the field of system identification, MCMC variants such as the one recently proposed by Green (2015) are mostly focused on offline system identification. This is partly due to computational challenges which prevent real-time use (Kuindersma et al., 2012). The standard MCMC algorithm is not suitable for model variation since different candidates do not share the same parameter set. Green (1995) first introduced reversible jump Markov chain Monte Carlo (RJMCMC) as a method to address the model selection problem. In this method, an extra pseudo random variable is defined to handle dimension mismatch. There are further extensions of MCMC in the literature, however, an online implication of it has yet to be reported.

Motion filtering and force prediction of robotic manipulators are important fields of study with interesting challenges suitable for Bayesian inferences to address (Saar et al., 2018). Here, measurements are inherently noisy, which is not desirable for control purposes. Likewise, inaccuracy, inaccessibility, and costs are typical challenges that make force measurement not ideal for practical use (Agand et al., 2016). Different environmental identification methods have been proposed in the literature

for linear and Gaussian noise (Wang et al., 2018); however, in cases of nonlinear models like Hunt-Crossley that does not have Gaussian noise (e.g. impulsive disorder), there is no optimal solution for the identification problem. Diolaiti et al. (2005) proposed a double-stage bootstrapped method for online identification of the Hunt-Crossley model, which is sensitive to parameter initial conditions. Carvalho & Martins (2019) proposed a method to determine the damping term in the Hunt-Crossley model. A neural network-based approach was introduced to control the contact/non-contact Hunt-Crossley model in (Bhasin et al., 2008)

This paper proposes a new technique, Adaptive Recursive Markov Chain Monte Carlo (ARMCMC), to address address certain weaknesses of traditional online identification methods, such as only being appllicable to systems Linear in Parameter (LIP), having Persistent Excitation (PE) requirements, and assuming Gaussian noise. ARMCMC is an online method that takes advantage of the previous posterior distribution, given that there is no sudden change in the parameter distribution. To achieve this, we define a new *variable jump distribution* that accounts for the degree of model mismatch using a *temporal forgetting factor*. The temporal forgetting factor is computed from a model mismatch index and determines whether ARMCMC employs modification or reinforcement to either restart or refine parameter distribution. As this factor is a function of the observed data rather than a simple user-defined constant, it can effectively adapt to the underlying dynamics of the system. We demonstrate our method using two different examples: soft bending actuator and Hunt-Crossley model and show favorable performance compared to state-of-the-art baselines.

The rest of this paper is organized as follows: In Sec. 2, introductory context about the Bayesian approach and MCMC is presented. Sec 3 is devoted to presenting the proposed ARMCMC approach in a step-by-step algorithm. Simulation results on a soft bending actuator with empirical results on a reality-based model of a soft contact environment capturing a Hunt-Crossley dynamic are presented in Sec. 4. Lastly, the final remarks and future directions are concluded in Sec 5.

## 2 Preliminaries

### 2.1 Problem Statement

In the Bayesian paradigm, estimates of parameters are given in the form of the posterior probability density function (pdf); this pdf can be continuously updated as new data points are received. Consider the following general model:

$$Y = F(X, \theta) + \nu, \tag{1}$$

where $Y$, $X$, $\theta$, and $\nu$ are concurrent output, input, model parameters and noise vector, respectively. To calculate the posterior probability, the observed data along with a prior distribution are combined via Bayes' rule (Khatibisepehr et al., 2013). The data includes input/output data pairs $(X, Y)$. We will be applying updates to the posterior pdf using batches of data points; hence, it will be convenient to partition the data as follows:

$$D^t = \{(X, Y)_{t_m}, (X, Y)_{t_m+2}, \cdots, (X, Y)_{t_m+N_s+1}\}, \tag{2}$$

where $N_s = T_s/T$ is the number of data points in each data pack with $T, T_s$ being the data and algorithm sampling times, respectively. This partitioning is convenient for online applications, as $D^{t-1}$ should have been previously collected so that the algorithm can be executed from $t_m$ to $t_m + N_s+1$ or algorithm time step $t$. Ultimately, inferences are completed at $t_m+N_s+2$. Fig. 1 illustrates the timeline for the data and the algorithm. It is worth mentioning that the computation can be done in parallel by rendering the task of the adjacent algorithm step (e.g. phase A of algorithm $t$, phase B of algorithm $t-1$ and phase C of algorithm $t-2$ can all be done simultaneously) According to Bayes' rule and assuming data points are independent and identically distributed ( i.i.d) in equation 1, we have

$$P(\theta^t | [D^{t-1}, D^t]) = \frac{P(D^t | \theta^t, D^{t-1}) P(\theta^t | D^{t-1})}{\int P(D^1 | \theta^t, D^{t-1}) P(\theta^t | D^{t-1}) d\theta^t}, \tag{3}$$

where $\theta^t$ denotes the parameters at current time step. $P(\theta^t | D^{t-1})$ is the prior distribution over parameters, which is also the posterior distribution at the previous algorithm sampling time. $P(D^t | \theta^t, D^{t-1})$ is the likelihood function which is obtained by the one-step-ahead prediction:

$$\hat{Y}^{t|t-1} = F(D^{t-1}, \theta^t), \tag{4}$$

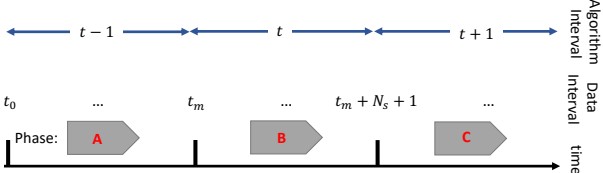

Figure 1: Timeline for data and different phase of ARMCMC algorithm. For algorithm at time t: Phase (A) Data collection [pack $N_s$ data points], Phase (B) Running [apply the method on the data pack], (C) Execution [update the algorithm results on parameters].

where $\hat{Y}^{t|t-1}$ is the prediction of the output in (1). If the model in (4) is valid, then the difference between the real output and predicted should be measurement noise, (i.e., $Y^{t|t-1} - \hat{Y}^{t|t-1} = \nu$). Therefore, the model parameter may be updated as follows:

$$P\big(D^t|\theta^t, D^{t-1}\big) = \prod_{t_m+1}^{t_m+N_s+1} P_\nu\big(Y^{t|t-1} - \hat{Y}^{t|t-1}\big), \tag{5}$$

where $P_\nu$ is the probability distribution of noise. Note that there is no restriction on the type of noise probability distribution.

*Remark 1:* As it was mentioned before, there is no need to know the exact probability distribution of noise. This probability distribution can be simply substituted with a Gaussian distribution, if one has minimal knowledge of the mean and variance of the data which can be easily obtained with preprocessing (Bishop, 2006).

## 2.2 MARKOV CHAIN MONTE CARLO

MCMC is often employed to compute the posterior pdf numerically. The multidimensional integral in (3) is approximated by samples drawn from the posterior pdf. The samples are first drawn from a different distribution called proposal distribution, denoted $q(.)$, which can be sampled easier compared to the posterior. Brooks et al. (2011) discuss different types of MCMC implementations which may employ various proposal distributions and corresponding acceptance criteria. The main steps of the Metropolis-Hastings algorithm are listed as follows (Ninness & Henriksen, 2010):

1. Set initial guess $\theta_0$ while $P(\theta_0|Y) > 0$ for iteration $k = 1$,
2. Draw candidate parameter $\theta_{cnd}$, at iteration $k$, from the proposal distribution, $q(\theta_{cnd}|\theta_{k-1})$
3. Compute the acceptance probability,

$$\alpha(\theta_{cnd}|\theta_{k-1}) = \min\left\{1, \frac{P(\theta_{cnd}|D)q(\theta_{k-1}|\theta_{cnd})}{P(\theta_{k-1}|D)q(\theta_{cnd}|\theta_{k-1})}\right\}, \tag{6}$$

4. Generate a uniform random number $\gamma$ in $[0, 1]$,
5. 'Accept' candidate if $\gamma \leq \alpha$ and 'ignore' it if $\gamma > \alpha$,
6. Set iteration to $k + 1$ and go to step 2.

## 2.3 PRECISION AND RELIABILITY

Two important notions in probabilistic framework to compare results are precision ($\epsilon$) and reliability ($\delta$). The former represents the proximity of a sample to the ground truth, and the latter represents the probability that an accepted sample lies within $\epsilon$ of the ground truth.

*Lemma:* Let $P_k$ be $k$ samples from MCMC, and $\mathbb{E}(P_k)$ denote its expected value. According to Chernoff bound (Tempo et al., 2012), given $\epsilon, \delta \in [0, 1]$, if the number of samples ($k$) satisfies

$$k \geq \frac{1}{2\epsilon^2} \log\big(\frac{2}{1-\delta}\big), \tag{7}$$

then $Pr\Big\{\{P_k - \mathbb{E}(P_k)\} \leq \epsilon\Big\} \geq \delta$.

---

**Algorithm 1** ARMCMC

---

*Assumptions:* 1) roughly noise mean ($\mu_\nu$) 2) roughly noise variance ($\sigma_\nu$) 3) desired precision and reliability ($\epsilon_0, \delta_0$) 4) desired threshold for model mismatch ($\zeta_{th}$)
*Goal:* Online calculation of parameters posterior distribution given the consecutive $t$-th pack of data ($P(\theta^t | D^t)$)
**Initialization:** Prior knowledge for $\theta_1^0$, n=0
Consider desire precision and reliability ($\epsilon, \delta$)
**repeat**
    Put $t_0 = n * N_s + 1$ from (2), n++
    Add new data pack to dataset $D^t$
    *Model mismatch index:* $\zeta^t$ from (10)
    **if** $\zeta^t < \zeta_{th}$ **then**
        *Reinforcement:* set prior knowledge equal to the latest posterior of previous pack
        *Temporal forgetting factor:* $\lambda^t$ from (9)
    **else**
        *Modification:* set prior knowledge $\theta_1^n$
        *Temporal forgetting factor:* $\lambda^t = 0$
    **end if**
    *Set minimum iteration $k_{min}$ from (12)*
    **for** $k = 1$ **to** $k_{max}$ **do**
        *Proposal distribution:*
            • draw $\lambda_k \sim U(0,1)$
            • *Variable jump distribution:* $q_k^t(.)$ from (8)
        Draw $\theta_k^{t*} \sim q_k^t(.)$
        *Acceptance rate:* $\alpha(.)$ from (6)
        Draw $\gamma \sim U(0,1)$
        **if** $\gamma \leq \alpha$ **then**
            'Accept' the proposal
        **end if**
    **end for**
    **Wait** to build $D_{t_0}^{t_m + N_s + 1}$ (algorithm sample time)
**until** No data is obtained

---

# 3 ARMCMC ALGORITHM

At each time interval, ARMCMC recursively estimates the posterior distribution by drawing samples. The number of samples drawn is constrained by the desired precision and reliability, and the real-time requirement. On the other hand, the maximum number of data points in each data pack, $N_s$, is limited by the frequency of model variation, and the minimum is confined by the shortest required time such that the algorithm is real-time. We propose a variable jump distribution that enables both enriching and exploring. This will necessitate the definition of the temporal forgetting factor as a measure to reflect current underlying dynamics of the data. In other words, this parameter will show the validity of the previous model for the current data. We also prove that ARMCMC achieves the same precision and reliability with fewer samples compared to the traditional MCMC. Algorithm 1 summarizes ARMCMC.

## 3.1 VARIABLE JUMP DISTRIBUTION

We propose a variable jump distribution (also known as a proposal distribution) to achieve faster convergence, thereby enabling real-time parameter estimation:

$$q_k^t(\theta^t | \theta_{k-1}^t) = \begin{cases} P(\theta^{t-1} | D^{t-1}) & \lambda_k \leq \lambda^t \\ N(\mu_D, \sigma_\nu) & \lambda_k > \lambda^t \end{cases}, \tag{8}$$

where $\theta_{k-1}^t$ is the $(k-1)$-th parameter sample which is given by the $t$-th data pack throughout the MCMC evaluation. Averaging the second half of this quantity will construct $\theta^t$. $P(\theta^{t-1} | D^{t-1})$ is the posterior distribution of the parameters at the previous algorithm time step, and $N(\mu_D, \sigma_\nu)$ is a Gaussian distribution with $\mu_D, \sigma_\nu$ computed using sample-based mean and variance of $D^{t-1}$.

The hyperparameter $\lambda^t$ (*temporal forgetting factor*), is an adaptive threshold for the $t$-th pack that takes inspiration from in the classical system identification terminology; it regulates how previous knowledge affects the posterior distribution. Smaller values of $\lambda^t$ intuitively means that there may be a large sudden change in $\theta$, and thus more exploration is needed. Conversely, larger values of $\lambda^t$ is appropriate when $\theta$ is changing slowly, and thus previous knowledge should be exploited. As more data is obtained, better precision and reliability can be achieved.

## 3.2 TEMPORAL FORGETTING FACTOR

Depending on whether the distribution of the parameter $\theta$ has changed significantly, a new sample can be drawn according to the *modification* or the *reinforcement* mode. Reinforcement is employed to make the identified probability distribution more precise when it is not undergoing sudden change. Modification is employed otherwise to re-identify the distribution "from scratch". Therefore, we define a *model mismatch index*, denoted $\zeta^t$, such that when it surpasses a predefined threshold ($\zeta^t > \zeta_{\text{th}}$), modification is applied. In other respects, if $\zeta^t \leq \zeta_{\text{th}}$, then $\zeta^t$ is used to determine $\lambda^t$ as follows:

$$\lambda^t = e^{-|\mu_\nu - \zeta^t|}, \tag{9}$$

where $\mu_\nu$ is an estimation of the noise mean, by calculation of the expected value in relation (1). Note that employing modification is equivalent to setting $\lambda^t = 0$. The model mismatch index $\zeta^t$ itself is calculated by averaging the errors of the previous model given the current data:

$$\zeta^t = 1/N_s \sum_{n=1}^{N_s} \left( y_n^t - \mathop{\mathbb{E}}_{\theta \in \theta^{t-1}} (F(D^t(n), \theta)) \right), \zeta^0 = \infty \tag{10}$$

*Remark 2:* The model mismatch index accounts for all sources of uncertainty in the system. To calculate $\zeta_{th}$, one needs to precalculate the persisting error between the designated modeled and measured data. In other words, $\zeta_{th}$ is basically an upper bound for the unmodeled dynamics, disturbances, noises, and any other source of uncertainty in the system.

*Remark 3:* To avoid numerical issues, the summation of probability logarithms are calculated. In addition, each pair in the algorithm time sample is weighted based on its temporal distance to concurrent time. Therefore Eq. (5) is modified as

$$\log \left( P(\cdot) \right) = \sum_{t_m+1}^{t_m+N_s+1} \log P_\nu(e^t),$$

$$e^t = \left( Y_n^t - F^t(D^{t-1}(n), \theta^t) \right) e^{-\rho(N_s - n)}, \tag{11}$$

where $\rho \in [0, 1]$ is a design parameter that reflects the volatility of the model parameters, and $e^t = [e_1^t, ..., e_n^t, ..., e_{N_s}^t]$. For systems with fast-paced parameters, $\rho$ should take larger values. We are trying to solve the Bayesian optimization problem that gives us the pdf of the model parameters ($\theta$), when given the data pairs in the presence of uncertainty.

## 3.3 MINIMUM REQUIRED EVALUATION

**Theorem 1.** *Let $\epsilon$ and $\delta$ be the desired precision and reliability. . Furthermore, it can be assumed that the initial sample has enough number of evaluations (as in (7)). To satisfy the inequality in Eq. (7), the minimum number of samples $k$ in ARMCMC is calculated using this implicit relation:*

$$k_{min} = \frac{1}{2\epsilon^2} \log(\frac{2}{\lambda^t(1-\delta) + 2(1-\lambda^t)e^{-2\epsilon^2(1-\lambda^t)k_{min}}}). \tag{12}$$

*Proof. Samples from previous pdf:* According to the variable jump distribution in (8), given $k$ samples, the expected number of samples drawn from the previous posterior probability ($P(\theta|D^t)$) is $\lambda^t k$. By assumption, the algorithm has already drawn at least $k$ samples in the previous algorithm time-step. Consequently, by (7), the expected number of samples with distances less than $\epsilon$ from $\mathbb{E}(P_k)$ drawn from a previous distribution is at least $\lambda k \delta$.

*Samples from Gaussian:* By (8), there are $k_0 = (1 - \lambda^t)k$ samples drawn in expectation. According to (13), we have $Pr\left\{\{P_k - \mathbb{E}(P_k)\} \leq \epsilon\right\} \geq \delta_0$, where $\delta_0$ is given by rearranging (7):

$$\delta_0 = 1 - 2e^{-2\epsilon^2 k_0}. \tag{13}$$

Thus, the expected number of samples with distances less than $\epsilon$ from $\mathbb{E}(P_k)$ are at least $\delta_0(1 - \lambda^t)k$.

*Overall reliability:* The total expected number of samples with distances less than $\epsilon$ from $\mathbb{E}(P_k)$ is the summation of the two parts mentioned above. Hence it is obtained through dividing by $k$:

$$\delta_1 = \frac{(\lambda^t k \delta) + (\delta_0(1 - \lambda^t)k)}{k} \tag{14}$$

Given the new obtained reliability, which is greater than the desired one, helps us decrease the number of evaluations. For the sake of illustration, Fig. 2 presents the minimum required number of evaluations with respect to $\lambda$ for different precisions and reliabilities. As it can be seen, the MCMC is equal to ARMCM if $\lambda$ is always set to one. The number of evaluations in ARMCMC mitigates as the validity of the previous model increases. $\square$

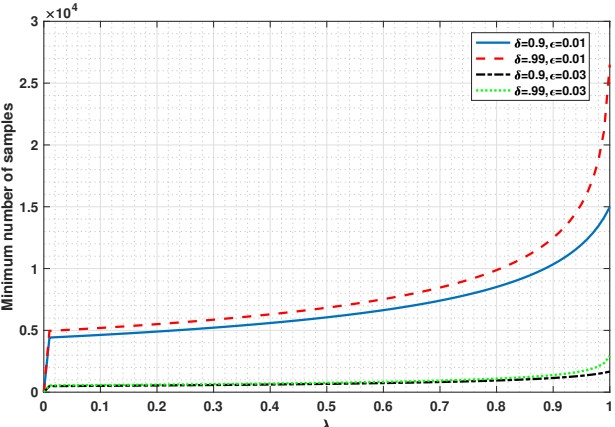

Figure 2: $K_{min}$ with respect to $\lambda$ for some values of $\epsilon, \delta$ in ARMCMC. (for $\lambda = 0$ evaluation for ARMCMC is equivalent to MCMC)

## 4 RESULTS

In this section, we demonstrate the priority of the proposed approach given two different examples. First, we employ the proposed method to identify the soft bending actuator model and compare the results with a Recursive Least Squares (RLS). In the second example, we evaluate it on the Hunt-Crossley model given reality-based data and compare it with a simple MCMC and RLS.

### 4.1 SIMULATION RESULTS

For this part, we consider the dynamic model of a fluid soft bending actuator. The dynamic is given by the following relation: (Wang et al., 2019)

$$\ddot{\alpha} = q_1(p - p_{atm}) - q_2\dot{\alpha} - q_3\alpha$$
$$u_c \operatorname{sign}(p_s - p)\sqrt{|p_s - p|} = q_4\dot{p} + q_5\dot{p}p, u_d = 0 \tag{15}$$
$$u_d \operatorname{sign}(p - p_{atm})\sqrt{|p - p_{atm}|} = q_6\dot{p} + q_7\dot{p}p, u_c = 0,$$

where $\alpha$ is the angle of the actuator, $u_c, u_d$ are the control inputs, and $p, p_s, p_{atm}$ are the current, compressor and atmosphere pressure respectively. For this example, we assume $q_1 = 1408.50, q_2 = 132.28, q_3 = 3319.40$ are known and $p_{atm} = 101.3kpa, p_s = 800kpa$. We are trying to identify the four other parameters $(q_4, ..., q_7)$. To this end, we assume the hybrid model below:

$$u \operatorname{sign}(p_s - p)\sqrt{|p_s - p|} = \theta_1\dot{p} + \theta_2\dot{p}p, u = \{u_c, u_d\} \tag{16}$$

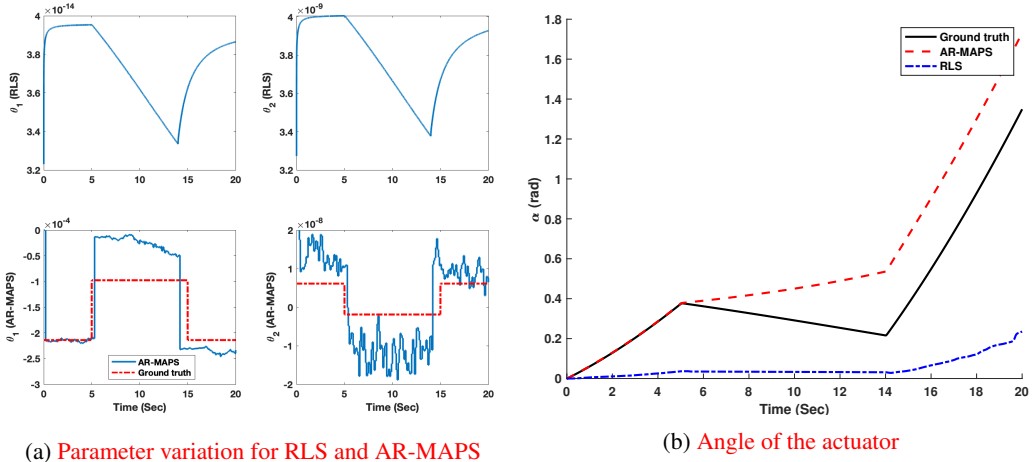

(a) Parameter variation for RLS and AR-MAPS

(b) Angle of the actuator

Figure 3: Comparison of RLS and AR-MAPS for soft bending actuator

As the range of these parameters are small, we scale the input vector by the factor of $10^7$ for RLS. Given the input $(u_c, u_d)$ and the output $(p, \dot{p})$, we want to identify the parameter and estimate the current angle of actuator knowing its initial position at the origin. The data sample time is $T = 1$ ms and each data pack includes 100 samples which results in an algorithm sample time equal to $T_s = 0.1$ sec. The point estimation obtained by considering the mode at the modification phase and the medium during the reinforcement phase in ARMCMC is denoted as AR-MAPS. The point estimate results for the parameter estimation are shown in Fig. 3a. The true parameters are $q_5 = -2.14 \times 10^{-4}, q_6 = 6.12 \times 10^{-9}, q_7 = -9.76 \times 10^{-5}, q_8 = -1.90 \times 10^{-9}$. The second norm of the parameters' errors are $0.0235, 6.0053 \times 10^{-7}$ for $\theta_1, \theta_2$ in RLS and $0.0089, 1.1840 \times 10^{-7}$ in AR-MAPS, respectively. Moreover, the estimation of the angle is plotted in Fig. 3b.

## 4.2 EMPIRICAL RESULTS

In this section, we demonstrate ARMCMC by identifying parameters of the Hunt-Crossley model, which represents an environment involving a needle contacting soft material. The needle is mounted as an end-effector on a high-precision translational robot, which switches between two modes: free motion and contact. Due to abrupt changes in the model parameters when the contact is established or lost, online estimation of the force is extremely challenging.

### 4.2.1 CONTACT DYNAMIC MODEL

Consider the dynamic of contact as described by the Hunt-Crossley model which is more consistent with the physics of contact than classical linear models such as Kelvin-Voigt (Haddadi & Hashtrudi-Zaad, 2012). In order to overcome the shortcomings of linear models, Hunt & Crossley (1975) proposed the following hybrid/nonlinear model :

$$f_e(x(t)) = \begin{cases} K_e x^p(t) + B_e x^p(t)\dot{x}(t) & x(t) \geq 0 \\ 0 & x(t) < 0, \end{cases} \tag{17}$$

in which $K_e, B_e x^p$ denote the nonlinear elastic and viscous force coefficients, respectively. The parameter $p$ is typically between 1 and 2, depending on the material and the geometric properties of contact. Also $x(t), \dot{x}(t), f_e$ are the current position, velocity (as inputs $X$) and contact force (as output ($Y$ in (1)) of a needle near or inside the soft material, with $x \geq 0$ representing the needle being inside. This needle can move freely in open space or penetrate the soft material; the forces on this needle are modeled using the Hunt-Crossley model. The practical problem we consider is to estimate the force at the tip of the needle by identifying the model parameters. $K_e, B_e, p$ are three unknown parameters ($\theta$ in Eq. (1)) that needs to be estimated. An online estimate of environment force plays a pivotal role in stable interaction between robotic manipulators and unknown environments.

To make the parameters ready for the regression problem we have

$$
\begin{aligned}
\log(f_e) &= \log(K_e x_s^p + B_e \dot{x}_s x_s^p), \\
\log(f_e) &= p \log(x_s) + \log(K_e + B_e \dot{x}_s).
\end{aligned}
\tag{18}
$$

We will use the RLS method proposed in (Haddadi & Hashtrudi-Zaad, 2012) as a baseline of comparison where it is needed to make the assumption that $B_e/K_e \dot{x}_s << 1$. It should be noticed that the vector of parameters $(\theta)$ in the following relation are not independent, which may lead to divergence. With this assumption, we have

$$
\begin{aligned}
\log(1 + B_e/K_e \dot{x}_s) &\approx B_e/K_e \dot{x}_s, \\
\log(f_e) &= p \log(x_s) + \log(K_e) + B_e/K_e \dot{x}_s.
\end{aligned}
\tag{19}
$$

$$
\begin{aligned}
\phi &= [1, \dot{x}_s, ln(x_s)], \\
\theta &= [\log(K_e), B_e/K_e, p]^T.
\end{aligned}
\tag{20}
$$

### 4.2.2 Setup

The data structure is same as previous simulation. Prior knowledge of all three parameters $(K_e, B_e, p)$ are initialized to $N(1, 0.1)$ (a normal distribution with $\mu = 1$ and $\sigma = 0.1$) Moreover, more data is collected, the spread of the posterior pdf decreases. A bit after 5 seconds, the needle goes outside of the soft material, and experiences zero force; this is equivalent to all parameters being set to zero. The color-based visualization of probability distribution over time is used for the three parameters in Fig. 4a. During the period of time that the whole space is blue (zero probability density), there is no contact and the parameter values are equal to zero.

Since we are taking a Bayesian approach, we are able to estimate the entire posterior pdf. However, for the sake of illustration, the point estimates are computed by using AR-MAPS method which is shown in Fig. 4b for the time-varying parameters $\theta_1 = K_e, \theta_2 = B_e, \theta_3 = p$. During the times that RLS results are chattering due to the use of saturation (if not, the results would have diverged), the needle is transitioning from being inside the soft material to the outside or vice versa. In addition, due to the assumption(19), performance can deteriorate even when there is no mode transition. Furthermore, in the RLS approach, estimated parameters suddenly diverged during free motion, since the regression vectors are linear dependent. In contrast, with the Bayesian approach, this issue can be easily resolved. The result of ARMCMC is presented in Fig. 5, which shows the force estimation with two different identification approaches. This probability of interest can be easily obtained by deriving the parameter density at one's disposal.

### 4.2.3 Quantitative Comparison

Quantitative details of comparing a naive point estimate of the ARMCMC algorithm by averaging the particles (AR-APS) and the RLS method are listed in Table 1. This reveals more than a 70% improvement in the precision of all model parameters throughout the study by using the Mean Absolute Error (MAE) criteria and also more than a 55% improvements in the second norm of force estimation error. Among parameters, the viscose $(B_e)$ has the largest error in the RLS method since it is underestimated due to the restrictive assumption in Eq. (19). The AR-MAPS approach uplifts the performance of the parameter identification and the force estimation.

We also compare ARMCMC to MCMC. For the algorithm to run in real-time, MCMC requires more time to converge. For this example, with $\lambda = 0.7$, the value of $K_{min}$ is 15000 for MCMC but only 6000 for ARMCMC (more than two times faster) with $\epsilon = 0.01, \delta = 0.9$. Two approaches that can be used to fix this drawback are to reduce the number of samples, which results in worse precision and reliability compared to ARMCMC, or to increase the algorithm sample time which would cause more delay in the estimation result and slower responses to changes in the parameter.

## 5 Conclusions

This paper presented an algorithm for an online identification of full probability distribution of model parameters in a Bayesian paradigm using an adaptive recursive MCMC. Due to the abrupt

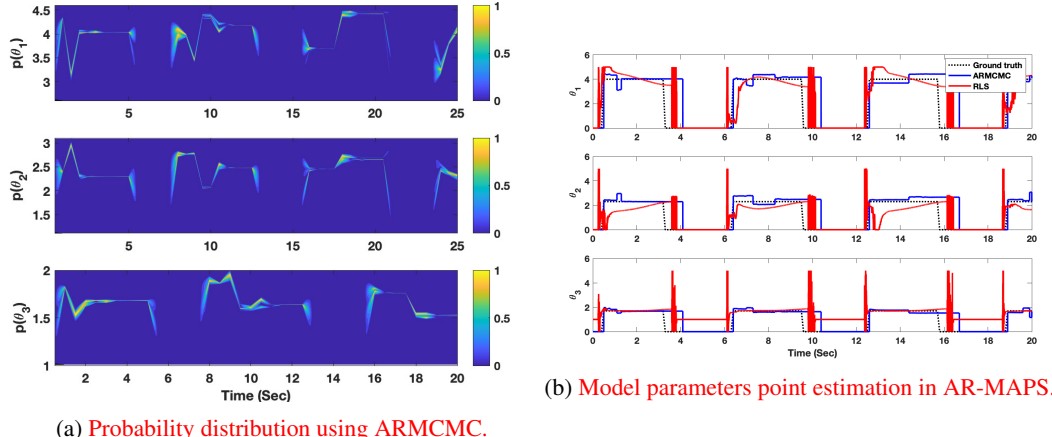

(a) Probability distribution using ARMCMC.

(b) Model parameters point estimation in AR-MAPS.

Figure 4: Estimation of model parameters ($\theta_1 = K_e, \theta_2 = B_e, \theta_3 = p$) for Hunt-Crossley

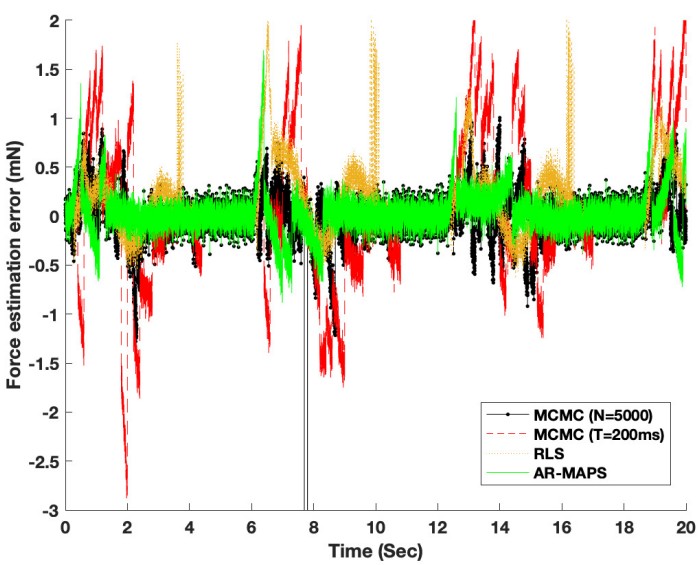

Figure 5: Force prediction error in RLS, AR-APS, and MCMC

Table 1: Comparison of RLS Haddadi & Hashtrudi-Zaad (2012) and point estimate of ARMCMC and MCMC for environment identification.

| ERRORS | MAE $K_e$ | MAE $B_e$ | MAE $p$ | RMS $F_e$ (mN) |
|---|---|---|---|---|
| RLS | 0.5793 | 0.9642 | 0.3124 | 51.745 |
| MCMC (N=5000) | 0.6846 | 0.8392 | 0.3783 | 76.695 |
| MCMC ($T_s = 0.2$) | 0.7294 | 0.9964 | 0.4195 | 101.88 |
| AR-APS | 0.0774 | 0.0347 | 0.0945 | 33.774 |
| AR-MAPS | 0.0617 | 0.0316 | 0.0756 | 31.659 |

change of model parameters such as contact with a soft environment when it is established/lost, conventional approaches suffer from low performance. Empirical results on the Hunt-Crossley model as a nonlinear hybrid dynamic model was compared with a well-known conventional identification process and revealed proficiency of the proposed algorithm. The proposed method provides a systematic strategy for handling abrupt changes which relaxes the pre-requirement conditions in the parameters. As future work, deploying a fully probabilistic framework from identification to control and a decision-making stage is considered to exploit the full potentials of the Bayesian optimization. Additionally, employing a method to compensate the delay will be taken into consideration.

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

## A    APPENDIX

According to Abolhassani et al. (2007), a nonlinear hybrid model based on a reality-based soft environment is considered as follows:

$$f_e = f_{\text{st}}(x, t, t_p) + f_{\text{fr}}(x, \dot{x}) + f_{\text{ct}}(x, t, t_p), \tag{21}$$

where $x$ is the needle tip position and $t_p$ is the latest time of puncture. Initial position of the environment is assumed to be at the origin. The stiffness of the force ($f_{\text{st}}$) belongs to a pre-puncture and the friction ($f_{\text{fr}}$) and cutting forces ($f_{\text{ct}}$) belong to a post-puncture. The stiffness force is modeled using a nonlinear Hunt-Crossley model:

$$f_{st}(x, t, t_p) = \begin{cases} 0 & x < 0 \\ K_e x^p(t) & 0 \leq x \leq x_1, t < t_p \\ 0 & x > x_2, t \geq t_p \end{cases} \tag{22}$$

where $K_e, p$ are the same parameters defined in (17). The maximum depth that the soft environment yields before the puncture and its position after it is denoted by $x_1, x_2$, respectively ($0 < x_2 < x_1$). In this study, the needle can insert up to 16.65, 10.21 $mm$ before and after penetration. A friction model is inspired from modified Karnopp model.

$$f_{fr}(x, \dot{x}) = \begin{cases} C_n sgn(\dot{x}) + B_e x^p \dot{x} & \dot{x} \leq -\Delta v/2 \\ \max(D_n, F_a) & -\Delta v/2 < \dot{x} \leq 0 \\ \max(D_p, F_a) & 0 < \dot{x} < \Delta v/2 \\ C_p sgn(\dot{x}) + B_e x^p \dot{x} & \dot{x} \geq \Delta v/2 \end{cases} \tag{23}$$

where $C_n = -11.96 \times 10^{-3}$ and $C_p = 10.57 \times 10^{-3}$ are negative and positive values of dynamic friction, $D_n = -0.01823$ and $D_p = 0.01845$ are negative and positive values of static friction, and $B_e, p$ are same as Eq. (17). The relative velocity between the needle and tissue is denoted by $\dot{x}$, and $\Delta v/2 = 0.005$ is the value below where the velocity is considered to be zero, and $F_a$ is the sum of non-frictional applied forces. The cutting force is considered as a static force constant for the tissue and the needle geometry if soft tissues are not inhomogeneous and anisotropic Cowan et al. (2011)

$$f_{ct}(x, t, t_p) = \begin{cases} 0 & x \leq x_1, t < t_p \\ 0.94 & x > x_2, t \geq t_p \end{cases}. \tag{24}$$

According to the previous relations, the system is considered as a hybrid model while providing both free motion and in-contact environment. The manipulator is a translational mechanism with a friction, slip, and hysteresis loop for the actuator. To present the superiority of the proposed algorithm, the results are compared with the RLS method presented in (Haddadi & Hashtrudi-Zaad, 2012). To prevent the results of RLS from divergence in model mismatch sequences, saturation is applied in the outputs of the identifier.

