# OpenReview forum: "ARMCMC: Online Model Parameters full probability Estimation in Bayesian Paradigm"
_ICLR.cc/2021/Conference — Reject_

### Official Review · AnonReviewer2 · 2020-10-27
**Interesting paper with good results, but related work seems overlooked**

**Rating:** 6
**Confidence:** 4

**Review:**

**The changes of the review after rebuttal are  indicated in bold text.**

### Summary of the contribution
This article proposes an algorithm called ARMCMC (Adaptive Recursive Markov Chain Monte Carlo" in the context of contact dynamic, for example when detecting the presence/absence of contact over time between a needle and a soft material.
ARMCMC aims at estimating the parameters of the Hunt-Crossley model, related to the elasticity and the viscosity of the material.
The algorithm handles both a real time estimation, and the fact that there are sudden changes in the dynamic (contact/no contact).
After reminding the basics of Bayesian estimation and MCMC, Section 2 finishes with the description of the contact dynamic model. Section 3 describes the ARMCMC algorithm and its main components, while Section 4 applies it to the example cited and compares it to Recursive Least-Squares (RLS).

### Strengths
The paper is rather well written and interesting.
It tackles a problem that seems difficult, especially since it is discontinuous and it is treated in real time, hence the need for both accurate and fast estimation.

### Weaknesses and concerns
1. Organization of the paper and clarity
- The title and introduction are a bit misleading as we expect to see a general method, while it appears later that it is designed specifically for the example given in the experiment.
If it is more general than that, I encouraged the authors to discuss other possible applications.

**The authors added an experiment to show that the method is more general than it appeared in the first version, giving more strength to the paper.**

- For a submission to a conference in artificial intelligence, it would be better to have less details on the well known MCMC (or at least put the details in appendix),
and more details about the specific application.
- The application envisioned is only specified in Section 4, while it would improve the comprehension were it already described in Section 2.
In particular, there is mention of a needle after Eq 10, the purpose of which we only understand 3 pages later.
2. Related work
Only one other algorithm is compared with the one proposed (RLS) it seems to me that related work has been overlooked.
- please discuss how ARMCMC differs from RJMCMC (Reversible Jump MCMC), where we could jump between model 1 (no contact) and model 2 (contact) back and forth.

**The authors answered that comment a bit quickly, in my opinion, as RJMCMC has been applied to a change-point problem in the original paper, which seems to be similar to the one presented in the paper. However, their argument on realtime is valid.**

- also, how does the proposed algorithm relate to the following ones?
	o "Particle filters for non-Gaussian hunt-crossley model of environment in bilateral teleoperation", Agand et al (2016), which is cited for other reasons
	o "Neural Network Control of a Robot Interacting With an Uncertain Hunt-Crossley Viscoelastic Environment", Bhasin et al (2008)
	o "Exact restitution and generalizations for the Hunt–Crossley contact model", Carvalho and Martins (2019)

**The authors agree that the work mentioned propose different approaches to the same model, but do not compare these approaches to the one they proposed in their experiment, and do not give enough details as to why.**

3. Minimum number of evaluations
Figure 1 shows an evaluation of the minimum number of evaluations depending on the values of the parameters:
- how come there are values lower than 1?
- In practice, was $k$ round up from $k_{min}$?
- is it in purpose that it is called $k_{min}$ in Eq 17 and Figure 1, and $k_{max}$ in Algorithm 1?

**The authors answered all the questions related to $k_{min}$, and it is now clearer.**

4. Empirical results
- The setting of the experiment is not specified: which material was used? Or was it synthetic data?
- There is a mention of results without EM at the end of section 4.1, but are there any WITH EM, and if not why is it so?
- In Table 1, why use different metrics for the parameters and for $F_e$? Why not use RMS everywhere? Does that mean that there are some points that are strongly over- or under-estimated? Or is it because of the lag (see comment below)?
- In Figure 3:
	o how come the lag for the absence of contact is so important with ARMCMC, while there is almost no lag for the beginning of contact? is it because of the temporal factor?
	o Can you comment on why the RLS looks so unstable?
	o also, which of the 2 versions of ARMCMC is presented here? APS or MAPS?
	o Why is there no comparison with MCMC in Figure 3, whereas it appears in all other results?
- Figure 4 is rather hard to read: please consider using doted/dashed lines and possibly markers to distinguish better between the estimators
- There is mention in the introduction and in Section 2 that RLS requires some restrictive conditions:
	o are they not realistic in practice?
	o it would be interesting to see a case where those conditions are not met, showing the strength of the proposed algorithm

### Question
Do the parameters for viscosity and elasticity of a material vary in time? Is it really important to keep estimating them in real time, or can't they just be updated once in a while?

### Minor comments
- Eq. 8 and 9, please replace the implication sign by a sentence for better readibility.
- In section 2.4, 2nd line: put the reference of Haddadi and Hashtrudi-Zaad in parenthesis to improve readibility.
- After Eq.10, the third parameter is designed by $n$ instead of $p$.
- After Eq 11, the acronym RLS is not defined: I had to go see the reference to see that the R stands for recursive, and not reweighted e.g.
- Eq. 19 mentions the variables $\delta_2$ and $\delta_0$, but there are no $\delta_1$ anywhere.
- Please increase the fontsize of the axis and legends in Figures 2 to 4.
- **Typo "apllicable" on page 2**

### Overall evaluation
While the topic and the approach are interesting, I believe the state of the art has not been discussed sufficiently.
If some of the suggested work indeed relate to the proposed one, they should be included in the experiment for comparison.

**After rebuttal, I think the authors did a good job in clarifying some points and adding an example in the experiment, which is why I upgrade the score I gave. However, I don't understand why they don't compare in the experiment with some of the methods from the cited literature concerned with the same model, especially Diolaiti et al. (2005).**

---

> ### Author Response · Authors · 2020-11-24
> **Response to reviewer #3**
>
>
> -   **Comment:**
>
>     *Weaknesses and concerns* Organization of the paper ...
>
> -   **Authors’ Response:**
>
>     Thanks for the constructive comment. Due to the space limitation, we could not add the second example. However, in this revised version we decide to add a second comparison with a fluid soft bending actuator as another nonlinear/hybrid example.
>
> -   **Comment:**
>      1.  For a submission to a conference in ...
>
> -   **Authors’ Response:**
>
>     1.  Although we need to talk minimally on basic MCMC, it is still  required to declare some notations for consistency. Also, ARMCM  is an extension of MCMC which requires whole notation and parameters in hand. In the revised version, we try to provide more detailed explanation for the novel part.
>
>     2.  We devoted Sec 2.4 for contact dynamic model as preliminaries, and discuss it in more detail in Sec 4. We add a sentence in Sec 2.4 to bridge the gap between two sections.
>
>     3.  We add another test study to enrich our comparison.
>
>     4.  Thanks for the comment. RJMCMC tries to handle multiple model selection, however, ARMCMC tries to propose an realtime  implication proposing facilitator variables to determine the current stage of the underlying dynamic. A sentence added to the introduction of the revised paper.
>
>     5.  The first one (Agand et al, 2016) propose another approach to estimate/control the Hunt-Crossley model. Bhasin et al (2008) and Carvalho et al  (2019) are also added to the reference list to enrich bibliography .
>
>     6.  Nice catch. The figure was not cropped properly. The $y-axis$  has $\times 10^4$
>
>     7.  Yes, we choose the value of $K$ to be equal to $K_{min}$
>
>     8.  It was a typo on Algorithm 1. Thanks for noticing. $K_{max}$ is  limited according to hardware specification.
>
>     9.  The empirical results is based on a model-based dynamic of a  needle in contact to a soft tissue. Based on (Abolhassani et al., 2007) this model is very accurate. However, for the model, we only consider the Hunt-Crossley dynamic and all the other parts considered as unmodeled dynamic and uncertainty.
>
>     10. Due to the space limitation, we have not implement EM. So we decide to put it in future works.
>
>     11. RMS is the best measurement of error in Gaussian noise. In other
>         cases, as we have in this example, we try to show other metrics like MAE that show more improvement in error. That could be because of strong over/under estimation of RLS. However for force, since there are different source of uncertainty, we use RMS.
>
>     12. According to the timeline (Fig 1 of revised manuscript), there is always a two algorithm time step delay (here 0.2sec) for ARMCMC. The reason why absence of contact is more severe, is that even when there is no contact, due to measured noise or  disturbances, the Hunt-Crossley model is still valid and the  model mismatch index does not alter. However, if there is no force, as the needle reaches the soft contact, the forces grows and Hunt-Crossley model would be chosen.
>
>     13. There is no note for RLS in case of model change. Therefore, as the parameter converge, the covariance matrix in RLS starts to decrease, making the Kalman gain minimum. By abrupt change in the dynamic, the current covariance matrix can not handle this situation. An easy fix for this could be periodic covariance resetting, which will mitigate the overal performance of the RLS during normal cases.
>
>     14. All the figures and reported variables are for AR-MAPS. Just the corresponding row in Table 1 is AR-APS.
>
>     15. Because it does not have good performance and deteriorate the visual level of the figures. The figure would be too messy and
>         inconclusive.
>
>     16. Done
>
>     17. The Hunt-Crossley model and all hybrid systems are practical systems which does not satisfy the RLS restrictive assumptions.
>         Moreover, Gaussian noise is not always the case, as in impulsive noise.
>
>     18. If the RLS conditions satisfies, it would have better results compare to any other suboptimal approach. However, in case of
>         not LIP/Guassion,PE the results would degenerate, as we can see in the two proposed example.
>
> -   **Comment:**
>
> *Question* Do the parameters for viscosity ...
>
> -   **Authors’ Response:**
>
>     The answer to this
>     question is that technically they can change, by varying their chemical/mechanical behaviour. Meanwhile, the problem here is *abrupt* change of parameters (like for hybrid models) as slow changes can be targeted by simple RLS with forgetting factor. Yes, the solution is to update the model frequently, but what is the rate of this update and how to determine when to do so, are the questions which was answered in the paper.
>
> -   **Comment:**
> 1. Eq. 8 and 9, please replace ...
>
> -   **Authors’ Response:**
>
>     All done. Thanks for noticing.
>
> -   **Comment:**
>
> *Overall evaluation* While the topic ...
>
> -   **Authors’ Response:**
>
>   Thanks for the notice. We have add some related works to enrich our reference list.

---

### Official Review · AnonReviewer3 · 2020-10-28
**Nice, simple, novel contribution awfully explained**

**Rating:** 5
**Confidence:** 4

**Review:**

The paper introduces a new Markov chain Monte-Carlo (MCMC) algorithm to obtain and track the posterior distribution over unknown parameters in a non-linear system. Despite its simple elegance, i.e., the introduction of a data-driven _temporal forgetting factor_ into the usual Metropolis-Hastings algorithm, the approach is, to my knowledge, novel. Its discovery seems to be the result of the intersection between fields: system identification and Bayesian sampling techniques, leading to new bridges.

In fact, novelty and interdisciplinarity are the strong points of this paper, but mostly everything else is compromised due to a severe lack of clarity. Although Section 1 is clear and fulfills its role of contextualizing the paper well, from Section 2 onward the a) organization, b) notation, c) technical inaccuracies in writing, and d) lack of detail on the application make the paper hard to read. To give just one example in each category: a) a major change in the usual MCMC scheme is explained in (and only in) _Remark 2_, Section 2.1, which is supposed to be part of the preliminaries on MCMC, b) the number of different notations for the collected data is ludicrous, and they are impossible to distinguish, c) $P_k$ in $(9)$ is not a probability density function (PDF), simply a sample of the MCMC algorithm,  and d) Section 2.4 is confusing because $p$ and $n$ are sometimes interchanged, no contextualizing figure is provided, assumptions are not interpreted, etc.

I recommend to reject this paper, but only because I believe it should be rewritten to be exposed clearly. Although the contribution is worthwhile, the paper is confusing and misleading. I am open to changing my recommendation if the authors introduce major changes in the presentation, writing, and structure, while still presenting the same content.

Among writing inaccuracies, using the term "density estimation" in the title of the paper is among the worse. Any Bayesian approach obtains a posterior distribution (explicitly or implicitly), but this is not the same problem as estimating the probability distribution of some given data.

The following questions point to some of the key problems I have found in the paper, without any specific order:
* Have the authors evaluated the assumption that the Gaussian model for noise is sufficiently robust for the application even when other noise models are used for the empirical results?
* Could the authors define the times at which the algorithm is run and the times at which data is collected? as well as the relation between them? A simple, intuitive, and correct notation for the data is needed. That could be the first step towards obtaining it.
* Could the authors clarify in which sense the minimum number of data points is restricted (not confined) by the fact that the algorithm is aiming at real-time operation?
* Could the authors clarify the role of $\rho$? It is mentioned in _Remark 2_ and does not seem to appear again.
* Could the authors give some hints to the readers on how the expectation-maximization algorithm is used for delay compensation? This would greatly aid reproducibility.
* Could the authors explain what is, and discuss the results of, AR-MAPS in Table 1?
* Could the authors use clearer measures of improvement than percentages of improvement? These are ill-defined and it is hard to pin which percentages are being reported.
* Is the resulting posterior as informative when the parameters are zero because the needle is not in contact with the soft surface? Could the vertical axes in Figure 2 be expanded to explore that?
* Could the authors clarify in which sense their simulated data is realistic? The appendix is quite uninformative to non-experts in the application.

Suggestions to improve the paper -  not specifically related to my recommendation:
* Thoroughly revise the use of English: some sentences are missing key components.
* Emphasize and clarify throughout the paper the role of the _temporal forgetting factor_, and the fact that it is not simply a user-defined constant, but a function of the observed model mismatch: this can be confusing to those that are not used to the concept.
* Use the entire available horizontal space in Figures 2, 3 and 4: it will help visualize some of the key features and look better.
* Use transparency for the line plots in Figures 3 and 4: it will allow to see their individual behavior better.
* Resize the fonts in all plots to match the size of the Figure captions: this will make them much easier to read.
* Correct the references (capitalization, etc.), e.g., Hunt-Crossley vs hunt-crossley in (Haddadi, Hashtrudi-Zaad, 2012).

---

> ### Author Response · Authors · 2020-11-24
> **Response to reviewer #2**
>
> -   **Comment:**
>
> 1. a major change in the usual MCMC scheme is ...
>
> 2.  the number of different notations ...
>
> 3. $P_k$ in  (9)  is not a ...
>
> 4. Section 2.4 is confusing ...
>
> -   **Authors’ Response:**
>
>  Thanks for the comment. We try to address all issues throughout the paper, here are
>     some examples:
>
>     1.  Remark 2 is a usual trick to avoid numerical instability.
>         Considering the importance of comparing latest data pack with
>         the previous data pack, we originally presented this in the
>         preliminaries part. In the revision, we moved this content to
>         the algorithm section.
>
>     2.  Revised. A timeline is also added to make it more clear.
>
>     3.  Corrected.
>
>     4.  Corrected.
>
> -   **Comment:**
>
> The following questions point to some of the key problems I have found in the paper, without any specific order: ...
>
>
> -   **Authors’ Response:**
>
>  We have changed the title to “ARMCMC: Online Model Parameters full probability Estimation in Bayesian Paradigm”.
>
>     1.  It is true that according to central limit theorem, infinite sum
>         of different noise would be Gaussian; however, impulse noise is
>         a common source of uncertainty which can not be modeled as a
>         Gaussian. So the answer to this question would be, although
>         there are some practical examples in which Gaussian noise cannot
>         model the uncertainty in the system, it is still crucial to have
>         some general approach like ARMCMC for applications like online
>         force prediction for robotics manipulators interacting with an
>         environment.
>
>     2.  To answer this question, we decided to draw a timeline for the
>         algorithm/data interval in Fig. 1 of the revised manuscript.
>
>     3.  The minimum required evaluation is restricted to guarantee the
>         algorithm achieve the desire precision and reliability. While
>         its maximum is confined to assure the evaluation can be done in
>         the dedicated time slot due to hardware limitation. For the
>         first example, given
>         $\lambda = 0.7,\epsilon = 0.01, \delta = 0.9$ to compute the
>         minimum and hardware specification to compute maximum, we have
>
>         $$K_{min}=6000 \leq K \leq K_{max} \approx 8000$$
>
>     4.  $\rho$ reflects volatility of the model parameters. For systems
>         with fast-pacing parameters, $\rho$ should take larger values.
>         It has also added to the revised paper.
>
>     5.  Due to the space limitation, we have not implement that. So we
>         decide to put it in future works.
>
>     6.  This is directly from the paper: “The point estimation obtained
>         by considering the mode at modification phase and medium during
>         reinforcement in ARMCMC is denoted as AR-MAPS.” In other words,
>         when the probability distribution of parameters are not dense
>         enough (which would happen during modification phase), we choose
>         the mode other than medium. While as the parameter probability
>         get less dispersed, we choose the mean of particles as the point
>         estimate. The results in Table 1 shows AR-MAPS shows better
>         performance in point estimates while the underlying method for
>         this and AR-APS are exactly the same. That is the reason authors
>         think this can not necessarily a good measure to evaluate the
>         performance of ARMCMC. The real priority of this method would be
>         fulfilled when it is combined with probabilistic robust
>         approaches, to have a complete decision making on a
>         probabilistic era.
>
>     7.  MAE and RMS are the most standard ways to report the error. We
>         use percentage to present relative not absolute improvement. 70%
>         improvement is the minimum improvement among three parameters
>         and 55% is the improvement in the probability of interest (here
>         the force).
>
>     8.  Yes, As the system lose its contact with the environment, we put
>         the parameters equal to zero as an informative. However, we do
>         not know when the contact would be lost, this would be
>         calculated by computing the model mismatch index. We did not
>         expand that figure to show the zero, cause it would be less
>         noticeable to distinguish between other variables around the
>         true value.
>
>     9.  The empirical results is based on a model-based dynamic of a
>         needle in contact to a soft tissue. Based on (Abolhassani et
>         al., 2007), this model provide a meticulous model of the
>         contact. We use this to synthetic the data for simulation.
>         However, for the model, we only consider the Hunt-Crossley model
>         and all the other parts considered as unmodeled dynamic and
>         uncertainty.
>
> -   **Comment:**
>
> Suggestions to improve the paper - not specifically related to my recommendation: ...
>
> -   **Authors’ Response:**
>
>     1.  Done.
>
>     2.  Done.
>
>     3.  Done.
>
>     4.  Done.
>
>     5.  Done.
>
>     6.  Done.

---

### Official Review · AnonReviewer1 · 2020-11-02
**ARMCMC: ONLINE MODEL PARAMETERS DENSITY ESTIMATION IN BAYESIAN PARADIGM**

**Rating:** 7
**Confidence:** 4

**Review:**

**Quality**
Overall, the technical content appears correct. The authors present results that show their proposed ARMCMC method outperforms RLS and conventional MCMC for a Hunt-Crossley problem. The results also show the robustness of the proposed algorithm to adapt to abrupt changes in the model parameters.

**Clarity**
Overall, the paper is well written. Paper should be revised to improve clarity and correct typos (non-exhaustive):
•	Page 1: "Besides, inaccuracy, inaccessibility, and cost are the typical shortcomings that make them not ideal for solely use" What does "them" refer to.
•	Page 2: appllicable, performance, t_m in definition of D^t
•	Page 4: \sigma_e or \sigma_v?
•	Define \theta in Equation (1)
•	Equation (6): e^t vs e^t_n?
•	Page 6: k_{min} on both sides of (17)
•	Page 7: “\theta_1 = K_e; \theta_1 = B_e; \theta_3 = p”
•	Define all acronyms when first used (e.g., RLS).
•	Consistency in notation (ARMCMC-APS vs. AR-APS)
•	Captions should be more explanatory to describe figure/table content. Define acronyms in figure/table captions so that the reader does not rely on information in the text to interpret the information in the figure. Increase the font size in figures.
o	Figure 1: Scale on y-axis (fractional minimum number of samples?). Label x-axis.
o	Figures 2 and 3: Labels based on the actual parameters, not generic \theta_1, \theta_2, \theta_3. A better layout is to time align probability distribution (top) with the corresponding parameter estimates (bottom).
o	Figure 4: Plot the results separately to allow easier comparison

**Originality**
Authors provide a summary of the prior work and outline the shortcomings that their proposed work addresses. The authors claim the proposed adaptive modifications to the MCMC provide the flexibility for exploitation or exploration based on the estimated model mismatch, updating the proposal distribution to better match the current data conditions, and selecting the number of iterations for parameter estimation based on a desired level of precision and reliability.

**Significance**
This paper describes an adaptive recursive algorithm to address the restrictive requirements of MCMC in Bayesian parameter estimation (“systems Linear in Parameter (LIP), having Persistent Excitation (PE) requirements, and assuming Gaussian noise.”). The proposed algorithm can be applicable to other latent parameter inference problems.
---------------------------------
**Pros**
•	The authors outline the proposed modifications, which includes some assumptions and derivations - the differences between MCMC and ARMCMC are highlighted.
•	The proposed algorithm is compared with other algorithms.
•	The authors provide code (however, should be better annotated).

**Cons**
•	It is difficult to assess the generality/utility of the proposed modifications (e.g., variable jump distribution) to other applications as the performance of the proposed method is only demonstrated in a specific case. A more comprehensive analysis is needed to assess the implications of the various approximations on model performance (e.g., model initialisation, substitution with a Gaussian distribution in (15), forgetting factor).
•	It is not clear how some of the parameters are chosen, e.g., threshold (\zeta_{th}).
•	Authors should first describe the methods of the experiment before presenting results to facilitate reproducibility.
•	Provide quantitative results for the claim: "For the algorithm to run realtime, MCMC requires more time to converge." (page 7).

---

> ### Author Response · Authors · 2020-11-24
> **Response to reviewer # 1**
>
> -   **Comment:**
>
> *Quality* Overall, the technical content appears correct. The authors present results that show their proposed ARMCMC method outperforms RLS and conventional MCMC for a Hunt-Crossley problem. The results also show the robustness of the proposed algorithm to adapt to abrupt changes in the model parameters.
>
> *Clarity* Overall, the paper is well written. Paper should be revised to improve clarity and correct typos (non-exhaustive):
> 1. Page 1: "Besides, inaccuracy, inaccessibility, and cost are the typical shortcomings that make them not ideal for solely use" What does "them" refer to.
>
> 2. Page 2: appllicable, performance, $t_m$ in definition of $D^t$
>
> 3. Page 4: $\sigma_e$ or $\sigma_v$?
>
> 4. Define $\theta$ in Equation (1)
>
> 5. Equation (6): $e^t$ vs $e^t_n$?
>
> 6. Page 6: $k_{min}$ on both sides of (17)
>
> 7. Page 7: "$\theta_1 = K_e; \theta_1 = B_e; \theta_3 = p$"
>
> 8. Define all acronyms when first used (e.g., RLS).
>
> 9. Consistency in notation (ARMCMC-APS vs. AR-APS)
>
> 10. Captions should be more explanatory to describe figure/table content. Define acronyms in figure/table captions so that the reader does not rely on information in the text to interpret the information in the figure. Increase the font size in figures.
>
> 11. Figure 1: Scale on y-axis (fractional minimum number of samples?). Label x-axis.
>
> 12. Figures 2 and 3: Labels based on the actual parameters, not generic $\theta_1, \theta_2, \theta_3$.
>
> 13. A better layout is to time align probability distribution (top) with the corresponding parameter estimates (bottom).
>
> 14.  Figure 4: Plot the results separately to allow easier comparison
>
>
> -   **Authors’ Response:**
>
>     Thanks for the comments, here are the responses:
>
>     1.  it refers to \`\`force measurement". The corresponding line has
>         been revised accordingly.
>
>     2.  As the data and algorithm time intervals are different, we use
>         another notation of time ($t_m$) for data sample: data time
>         steps $t_m:t_m+N_s+1$ is equivalent to algorithm time step $t$.
>         This has been addressed in the revised version, a timeline has
>         also included to make it more clear.
>
>     3.  $\sigma_\nu$. Thanks for noticing. It is corrected in the paper
>         too.
>
>     4.  Corrected
>
>     5.  Corrected
>
>     6.  Yes, as it was noted on the paper, it is an implicit relation to
>         compute the $k_{min}$. One needs to precompute it to run the
>         algorithm.
>
>     7.  Corrected
>
>     8.  Done.
>
>     9.  Done.
>
>     10. Done.
>
>     11. Thanks for noticing. The figure was cropped excessively. It is
>         the minimum number of evaluation.
>
>     12. Corrected.
>
>     13. Thanks for the suggestion. After some consideration, we decided
>         to keep the current layout.
>
>     14. Due to the space limitation, it is not possible to dedicate
>         separate figure for each one. Instead, we made the whole plot
>         bigger.
>
> -   **Comment:**
>
> *Pros:*
>
> The authors outline the proposed modifications, which includes some assumptions and derivations  the differences between MCMC and ARMCMC are highlighted.
>
> The proposed algorithm is compared with other algorithms.
>
> The authors provide code (however, should be better annotated).
>
>  *Cons:*
>
> 1. It is difficult to assess the generality/utility of the proposed modifications (e.g., variable jump distribution) to other applications as the performance of the proposed method is only demonstrated in a specific case. A more comprehensive analysis is needed to assess the implications of the various approximations on model performance (e.g., model initialisation, substitution with a Gaussian distribution in (15), forgetting factor).
>
> 2. It is not clear how some of the parameters are chosen, e.g., threshold ($\zeta_{th}$).
>
> 3. Authors should first describe the methods of the experiment before presenting results to facilitate reproducibility.
>
> 4. Provide quantitative results for the claim: "For the algorithm to run realtime, MCMC requires more time to converge." (page 7).
>
> -   **Authors’ Response:**
>
>     1.  Thanks for the comment. To prove the generalizability of the
>         proposed method, another simulation with a different dynamical
>         system is conducted along with a more comprehensive analysis is
>         conducted to address this concern.
>
>     2.  The model mismatch index is calculated with preprocessing on
>         data and the designated model. It is basically an upper bound
>         for the unmodeled dynamic/ disturbances, noises and any other
>         source of uncertainty in the system. We add a sentence to the
>         paper to make it more clear.
>
>     3.  Noted. Thanks
>
>     4.  For this example, with $\lambda=0.7$, the value of $K_{min}$ is
>         $15000$ for MCMC, but only $6000$ for ARMCMC (more than two
>         times faster) with $\epsilon=0.01,\delta=0.9$. We also add this
>         in the simulation part of the revised paper.

---

### Decision · Program_Chairs · 2021-01-07
**Final Decision**

**Decision:**

Reject

**Comment:**

Pros:
- The authors propose a novel method to perform MCMC in the condition where there is a distribution over models describing the data, rather than just a distribution over the parameters of a single consistent model (ie, in the theme of reversible jump MCMC). Sampling from the posterior of mixtures of parametric models is something current MCMC algorithms are very bad at, so this addresses an important need.
- Reviewers believed the proposed technique was novel and technically correct.
- The paper appears to build a bridge between the fields of system identification and Bayesian sampling techniques

Cons:
- Major concerns were raised about lack of clarity
- - From a *lightweight* read, I also had difficulty understanding what the paper was proposing, or even the precise problem it was tackling
- Experimental validation was limited, and missing baselines
- During discussion, the paper had no strong advocate